# Structure and Oxidation Resistance of Mo-Y-Zr-Si-B Coatings Deposited by DCMS and HIPIMS Methods Using Mosaic Targets

Alina D. Sytchenko [1,*] , Pavel A. Loginov [1] , Alla V. Nozhkina [2], Evgeny A. Levashov [1]
and Philipp V. Kiryukhantsev-Korneev [1,*]

1 Laboratory "In Situ Diagnosis of Structural Transformations" of the Scientific—Educational Center of SHS, National University of Science and Technology MISIS, 119049 Moscow, Russia
2 Research Institute of Natural, Synthetic Diamonds and Tools, 129110 Moscow, Russia
* Correspondence: alina-sytchenko@yandex.ru (A.D.S.); kiruhancev-korneev@yandex.ru (P.V.K.-K.); Tel.: +7-(495)-638-46-59 (A.D.S. & P.V.K.-K.)

**Abstract:** In this study, Mo-(Y,Zr)-Si-B coatings were obtained by direct current magnetron sputtering (DCMS) and high-power impulse magnetron sputtering (HIPIMS) using mosaic targets. The results showed that the addition of Y and Zr into the composition of Mo-Si-B coatings led to the suppression of columnar grain growth, a decrease in the crystallite size of h-MoSi$_2$ phase from ~50 to ~5 nm, and an increase in the amorphous to crystalline phases ratio Doping of the Mo-Si-B coating with Y and Zr promoted an increase in oxidation resistance at a temperature of 1000 °C. The introduction of yttrium into the composition of Mo-Si-B contributed to an increase in their crack resistance when heated to 1300 °C. High oxidation resistance of the coatings was provided by a defect-free SiO$_2$ + MoO$_3$ + Y$_2$O$_3$ surface layer. The transition from the DCMS mode to HIPIMS decreased the texture of the Mo-Si-B coatings. The use of an HIPIMS mode led to a decrease in the oxidation rate of Mo-(Y)-Si-B coatings at T = 1000 °C by 1.6–4.5 times compared to DCMS. In the case of Mo-Y-Si-B coatings, the use of HIPIMS led to a decrease of more than 50% in the thickness of the oxide layer at a temperature of 1300 °C.

**Keywords:** direct current magnetron sputtering; high-power impulse magnetron sputtering; MoSi$_2$-based coating; yttrium; structure; oxidation resistance

## 1. Introduction

Molybdenum disilicide is the leading material in the family of ultra-high temperature ceramics due to its high melting point (2050 °C) and thermal conductivity (53 W·m$^{-1}$·K$^{-1}$), low thermal expansion coefficient (8·10$^{-6}$ °C$^{-1}$), significant strength at temperatures above 1000 °C, and relatively high oxidation resistance (up to 1700 °C) [1,2]. It is well known that the high resistance of MoSi$_2$ to oxidation is due to the formation of a protective SiO$_2$ layer due to partial oxidation of Si. At the same time, undesirable oxidation of Mo occurs at a temperature of 500 °C [3,4]. This is primarily caused by cracking of the silicate layer. To equalize the coefficients of thermal expansion and increase the ability of the coating to self-heal, boron is introduced into the composition [5,6]. Boron-doped MoSi$_2$ coatings have an operating temperature 1100–1400 °C higher than unalloyed coatings. The introduction of hafnium and zirconium into the composition of Mo-Si-B coatings leads to a decrease in the size of the crystallites of the h-MoSi$_2$ phase by 1.5–2 times, and also contributes to the healing of defects and decreases the oxidation depth at a temperature of 1500 °C by 40–70% [7]. It is known that doping of silicide coatings with rare earth elements leads to an increase in their oxidation resistance [8–11]. The introduction of Y$_2$O$_3$ into MoSi$_2$ coatings obtained by plasma spraying contributes to an increase in oxidation resistance due to the formation of various yttrium silicates, such as Y$_2$SiO$_5$, Y$_2$Si$_2$O$_7$, and Y$_4$Si$_3$O$_{12}$; an increase

in the viscosity of the $SiO_2$ oxide film; and a slowdown in the penetration of reactive oxygen species into the interior [9]. In turn, alloying Mo-Si-B with Y and La increases the resistance to oxidation at 650–950 °C due to the formation of $Y_2O_3$ and $La_2O_3$ oxides in the borosilicate layer, which reduces its viscosity and prevents $MoO_3$ from evaporating [10–13]. Mo-Si-B coatings doped simultaneously with Ti and Y are highly resistant to gas corrosion at T = 1800–2100 °C [14].

In most cases, coatings based on $MoSi_2$ are deposited by cementation [15,16], as well as by a method that combines Mo deposition with subsequent cementation of Si and B [17,18]. The disadvantages of the method are increased surface roughness and defectiveness, uneven composition, discontinuity, and thickness variation in coatings [19]. At the same time, additional machining of products with exact tolerances may be required. A promising method for obtaining Mo-Si-B coatings is magnetron sputtering (DCMS). Among the advantages of this method are easy control of the composition, structure, and properties of the coatings; low concentration of defects; low roughness; high purity; no restrictions on the choice of substrate material; preservation of the product's geometry; and relatively high deposition rates. The use of high-power impulse mode magnetron sputtering (HIPIMS) provides additional opportunities for improving the characteristics of coatings [20–23]. The HIPIMS method, due to its higher power, provides a significant increase in plasma density from ~$10^{10}$ ion/cm$^3$ for DCMS to $10^{13}$–$10^{14}$ ion/cm$^3$ for HIPIMS [24]. In the case of HIPIMS, the sputtered atoms are intensely ionized during their passage through the plasma, and the stream consists predominantly of ions rather than atoms, as in the case of conventional DCMS. An increase in the ion/atom ratio in the flow, which is inherent in HIPIMS, leads to a significant increase in the adhesive strength of deposited coatings due to the formation of pseudodiffusion layers and the effects of ion implantation at the stage of preliminary etching of the substrate's surface [25].

The aim of this work is to study the effect of yttrium addition on the structure and oxidation resistance of Mo-(Zr)-Si-B coatings obtained by the DCMS and HIPIMS methods using mosaic targets.

## 2. Materials and Methods

Mo-Si-B, Mo-Y-Si-B, and Mo-Zr-Y-Si-B coatings were produced by direct current magnetron sputtering (DCMS) and high-power impulse magnetron sputtering (HIPIMS) (Table 1). The coatings were deposited on functionally graded targets with a lower layer of Mo and upper working layers of compositions 90%$MoSi_2$ + 10%MoB, 80% (90%$MoSi_2$ + 10%MoB) + 20%$ZrB_2$. The initial raw materials used to obtain targets include powders of PM-99.95 grade Mo (particle size of 2 ÷ 10 µm); silicon of KEF-4.5 grade (particle size of 2 ÷ 45 µm); zirconium of PZrK-1 grade (particle size of 10 ÷ 20 µm); and amorphous boron of B-99 A grade (average particle size of 0.2 µm). Reactive mixtures of elemental powders were prepared in a ball mill for 8 h using steel vials and milling bodies; the green mixture-to-milling bodies mass ratio was 1/6. The ceramics with intended compositions were synthesized using the method of self-propagating high-temperature synthesis. Combustion products were milled in a ball mill with hard metal balls (1:10 mixture-to-balls ratio) for 8 h to produce micron-sized ceramic powders $MoSi_2$–MoB and $MoSi_2$–MoB–$ZrB_2$. The sintering of ceramics was performed by hot pressing using the DSP-515 SA press (Dr. Fritsch Sondermaschinen GmbH, Fellbach, Germany). Ceramics were hot-pressed at a temperature of 1600 °C, a heating rate of 10 °C/min, 35 MPa pressure, and a 10 min dwelling time.

**Table 1.** Elemental composition (GDOES data), thickness, and growth rate of coatings.

| Coating | Mode | $S_Y$, cm$^2$ | Elemental Composition, at.% | | | | | Thickness, μm | Growth Rate, nm/min |
|---|---|---|---|---|---|---|---|---|---|
| | | | Mo | Si | B | Y | Zr | | |
| Mo-Si-B | DCMS | 0 | 24 | 68 | 8 | 0 | 0 | 7.3 | 183 |
| Mo-Si-B | HIPIMS | 0 | 22 | 70 | 8 | 0 | 0 | 3.5 | 88 |
| Mo-Y-Si-B | DCMS | 10 | 30 | 58 | 5 | 7 | 0 | 6.4 | 160 |
| Mo-Y-Si-B | HIPIMS | 10 | 24 | 62 | 7 | 7 | 0 | 6.2 | 155 |
| Mo-Zr-Y-Si-B | HIPIMS | 5 | 14 | 52 | 22 | 3 | 9 | 3.2 | 80 |

When applying Mo-Y-Si-B and Mo-Zr-Y-Si-B coatings, Y plates with areas of $S_Y$ = 5 or 10 cm$^2$ were placed in the erosion zone of the corresponding target with an area of ~60 cm$^2$. Coatings were deposited in a vacuum setup based on a UVN-2M evacuation system, in the working space of which there were two planar disk magnetrons, a slot-type ion source, and a substrate attachment/positioning system [7]. To implement the DCMS mode, a Pinnacle Plus Advanced Energy power supply was used. In this case, a power of 1 kW was applied to the magnetron. High-power pulsed sputtering was carried out using a TruPlasma 4002 Trumpf system, while the average power was maintained at the same level of $1.0 \pm 0.3$ kW and the peak power reached 75 kW. Argon (99.9995%) was used as the working gas, the flow rate of which was 37.5 mL/min. The residual and operating pressures were $3 \times 10^{-3}$ and $1 \times 10^{-1}$ Pa, respectively.

Plates made of polycrystalline aluminum oxide, grade VK-100-1, were used as substrates. The substrates were ultrasonically cleaned in isopropyl alcohol on a UZDN-2T unit for 5 min. Immediately prior to coating deposition, the substrates were etched with Ar+ ions in a vacuum chamber using an ion source at an accelerating voltage of 2.5 kV for 10 min. The settling time was 40 min.

The structure and composition of the coatings were studied using scanning electron microscopy (SEM) and energy dispersive analysis (EDS) with an S-3400 microscope (Hitachi, Tokyo, Japan) equipped with a Noran System 7 attachment (Thermo Fisher Scientific, Waltham, MA, USA). Elemental profiles were obtained using a Profiler-2 (HORIBA Jobin Yvon) glow discharge optical emission spectrometer (GDOES) [26]. The fine structure of the coatings was studied by transmission electron microscopy (TEM) using a JEM-2100 (Jeol, Tokyo, Japan) microscope. Fourier transform (FFT) and calculation of interplanar distances were carried out for phase identification using Olympus Radius and ImageJ software. The study of the elemental composition at certain points and the mapping were carried out by the EDS method using an X-Max80 T detector (Oxford Inst., Abingdon, UK). X-ray diffraction (XRD) was carried out on a D2 Phaser diffractometer (Bruker, Karlsruhe, Germany) using CuKα radiation. X-ray photoelectron spectroscopy (XPS) studies were carried out on a Kratos instrument (Shimadzu, Tokyo, Japan). The excitation source was monochromatized Al Kα radiation (hν = 1486.6 eV) and the power was 50 W. To remove contaminants, the coatings were etched using an argon cluster ion source in the Ar500+ cluster mode at a voltage of 20 kV for 5 min. Residual stresses in the coatings were evaluated by measuring the curvature of the substrate before and after deposition of the coating using the Stoney formula [27]. Curvature measurements were performed using a WYKO-NT1100 (Veeco, New York, NY, USA) optical profilometer. To study the oxidation kinetics, the coatings were annealed in air in an SNOL-7.2/1200 muffle furnace at a temperature of 1000 °C and holding times of 10, 30, 60, and 180 min. To determine the resistance of the coatings to sudden cooling, quenching from a temperature of 1000 °C into water was carried out. High-temperature annealings at temperatures of 1300 and 1500 °C were carried out using a TK 15.1800 DM.1F (Termokeramika, Moscow, Russia) furnace. The sample heating rate and exposure were 15 °C/min and 60 min, respectively. After annealing, the coatings were studied by SEM, EDS, XRD, and TEM. Prior to TEM studies, fragments of the surface layers of the annealed coatings were mechanically removed from the substrates and placed on a copper grid holder.

## 3. Results and Discussion

### 3.1. Coating Structure

According to the GDOES data, all elements showed uniform depth distribution in the coatings. The oxygen impurity concentration in the coatings did not exceed 5 at.%. The averaged chemical composition of the Mo-(Y,Zr)-Si-B coatings is presented in Table 1. The Mo-Si-B coatings obtained by DCMS and HIPIMS had a similar composition; the content of the main elements was 22–24 at.% Mo, 68–70 at.% Si, and 8 at.% B. For the base coatings, some excess of silicon was observed compared to the stoichiometric composition: the Mo:Si ratio = 1:3. For the Mo-Y-Si-B DCMS coating, a yttrium concentration of 7 at.%, an increase in the Mo content to 30 at.%, and a decrease in the content of Si to 58 at.% and B to 5 at.% were observed. DCMS-coating Mo-Y-Si-B had a stoichiometric composition, with a Mo:Si ratio = 1:2.

When switching to the HIPIMS mode, the content of molybdenum decreased by 20%, while silicon and boron increased by 7 and 40%, respectively. This effect is associated with a high degree of ionization of the sprayed material's flow during coating deposition in the HIPIMS mode [28,29]. The Mo-Zr-Y-Si-B coating obtained using a target containing an additional 20% ZrB$_2$ had a high boron content of 22 at.%. The concentrations of Mo, Si, and Y were 14, 52, and 3 at.%, respectively.

According to cross-section SEM images (Figure 1), all coatings had a dense, low-defect structure. Based on the obtained data and the experience of previous works [5,7], the DCMS Mo-Si-B coating was characterized by a columnar structure. For the Mo-Si-B HIPIMS coating and alloyed samples, individual structural components were not detected at transverse fractures. The DCMS Mo-Si-B coating was characterized by a maximum thickness of ~7.3 µm and a growth rate of ~183 nm/min (Table 1). When a HIPIMS mode was applied, the thickness decreased by 2.1 times. This decrease in the growth rate may have been due to the self-sputtering effect, as well as to a decrease in the total sputtering time in the HIPIMS mode [30,31]. Mo-Y-Si-B coatings obtained by the DCMS and HIPIMS methods were characterized by close growth rates in the range of 155–160 nm/min. The Mo-Zr-Y-Si-B sample had a minimum thickness of ~3.2 µm and a growth rate of ~80 nm/min.

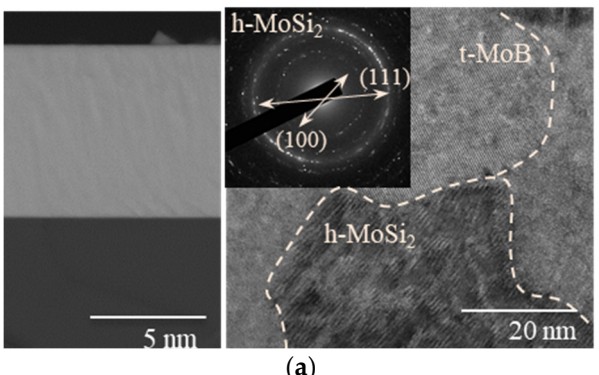

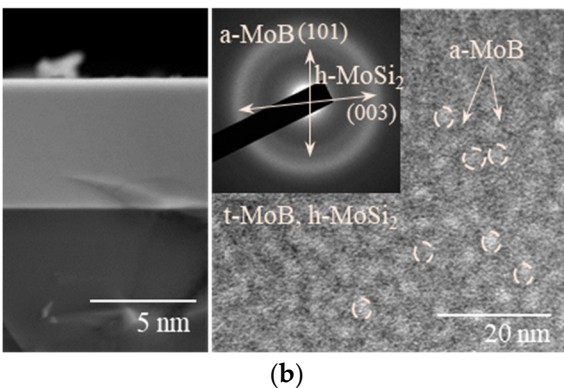

(**a**)                                                    (**b**)

**Figure 1.** Typical cross-section SEM images, selected area electron diffraction, and bright-field TEM images of Mo-Si-B (**a**) and Mo-Y-Si-B (**b**) coatings obtained by DCMS.

According to TEM micrographs, the Mo-Si-B coatings were characterized by a crystalline structure with a grain size of about 50 nm (Figure 1a). The electron diffraction patterns of the Mo-Si-B coating revealed the main reflections to have interplanar spacings of 0.392, 0.334, 0.247, and 0.215 nm, corresponding to the values of d/n = 0.398, 0.340, 0.253, and 0.217 nm of lines (100), (101), (102), and (111) of the h-MoSi$_2$ hexagonal phase (ICDD 80-4771). The analysis of high-resolution bright-field images revealed the presence of grains with interplanar spacings of 0.421 and 0.254 nm, close to the values for the t-MoB (ICDD 65-2753) and h-MoSi$_2$ phases, respectively. In contrast, the Mo-Y-Si-B coatings were characterized by a nanocomposite structure (Figure 1b). According to TEM images, the

coating contained rounded grains about 2–4 nm in diameter, surrounded by amorphous interlayers 1–3 nm thick. On the electron diffraction patterns from the Mo-Y-Si-B coating, a wide ring was revealed and two ring signals were observed. The observed halo ring pattern indicated the presence of a highly disordered phase. The interplanar spacing, determined from the middle of the halo's reflection, was 0.305 nm, which can be assigned to the (101) plane of the t-MoB tetragonal phase. In addition to the more intensive halo, a ring reflection with an interplanar spacing of 0.222 nm was observed, corresponding to the (003) plane of the h-MoSi$_2$ phase. According to high-resolution TEM images, fringe contrast with an interplanar spacing of 0.305 nm was also found for the (101) plane of t-MoB.

XRD patterns of the Mo-(Y,Zr)-Si-B coatings obtained by DCMS and HIPIMS are shown in Figure 2. For all coatings, peaks from the Al$_2$O$_3$ substrate (ICDD 10-0173) were detected. The XRD pattern of the Mo-Si-B coating showed high-intensity h-MoSi$_2$ peaks with textures in the (110) direction. The h-MoSi$_2$ grain size determined using the Scherrer equation from the (110) peaks was 53 nm. In the case of the Mo-Si-B HIPIMS coating, no noticeable texture of the h-MoSi$_2$ phase was found: the main high-intensity reflections associated with reflection from the (100), (110), and (200) planes were observed in the XRD pattern. The transition to the HIPIMS mode did not affect the size of the h-MoSi$_2$ crystallites.

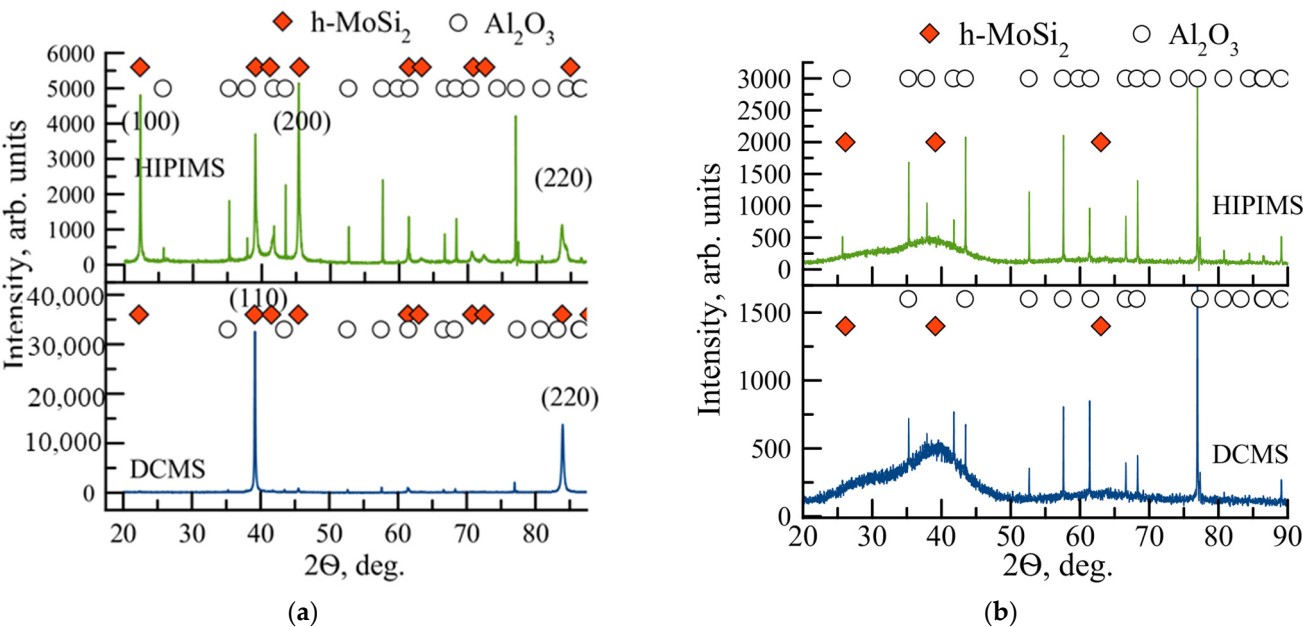

(a)  (b)

**Figure 2.** XRD patterns of Mo-Si-B (**a**) and Mo-Si-B-Y (**b**) coatings obtained by the DCMS and HIPIMS methods.

The h-MoSi$_2$ lattice parameters for the Mo-Si-B coatings did not differ from the values obtained for the h-MoSi$_2$ powder standard (a = 0.460 and c = 0.657 nm). On the XRD patterns of the Mo-Y-Si-B coatings obtained by the DCMS and HIPIMS methods, as well as the HIPIMS-coating of Mo-Zr-Y-Si-B, broadened peaks were revealed at positions 2θ = 24–34, 35–45, and 56–69°, which can be interpreted as peaks from the amorphous phase. Thus, the introduction of dopants (Y,Zr) contributed to the amorphization of the base Mo-Si-B coating, and the transition from the DCMS mode to HIPIMS reduced its texture.

For a better understanding of the chemical arrangement and bonding nature of Y-doped coatings, the XPS method was utilized. Figure 3 shows the high-resolution Mo3d, B1s, Y3d, and Si2p spectra taken from the Mo-Y-Si-B DCMS coating.

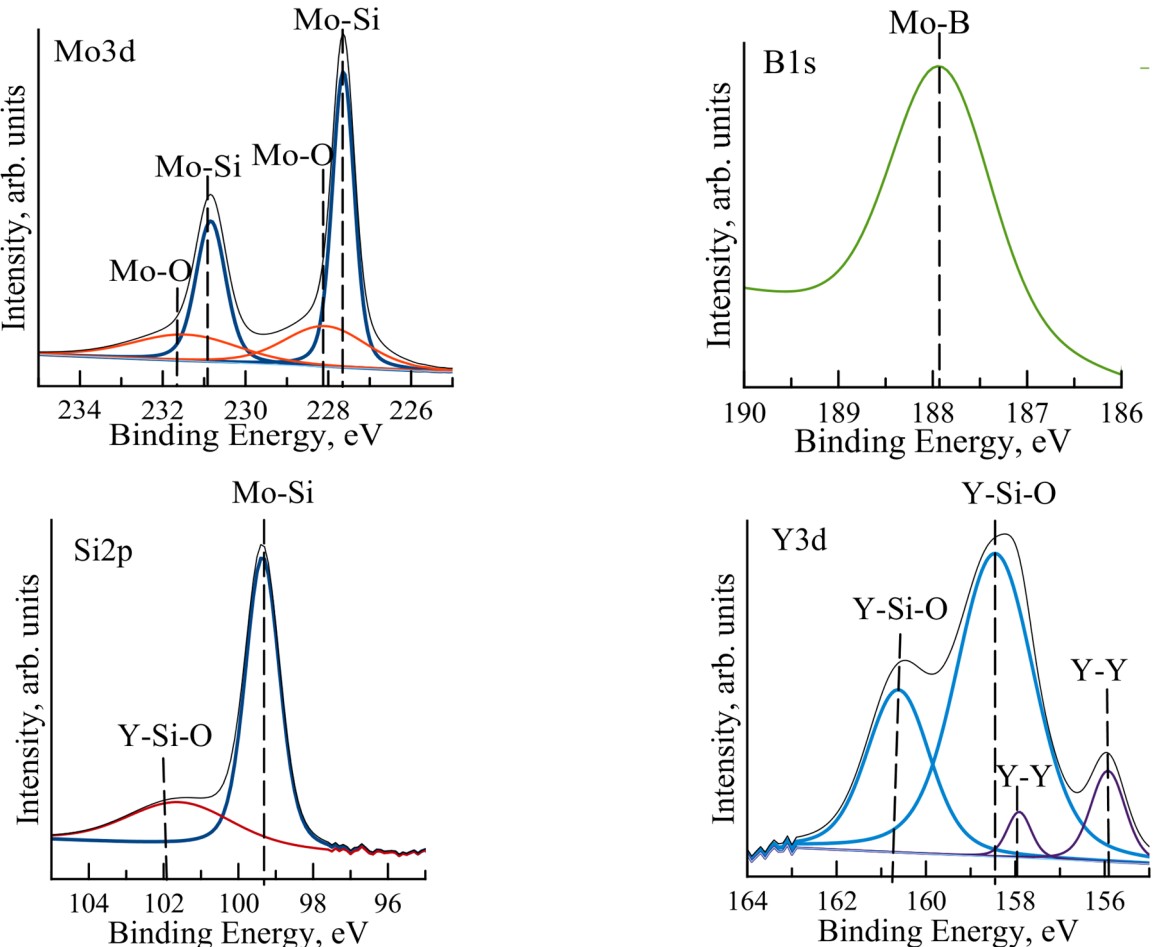

**Figure 3.** High-resolution Mo3d, B1s, Y3d, and Si2p XPS spectra for DCMS coating Mo-Y-Si-B.

For all coatings, the Mo3d spectrum contained two main doublets, described by the $3d_{5/2}$ and $3d_{3/2}$ states. Peaks corresponding to binding energies of 227.6 and 230.8 eV indicated that molybdenum was chemically bonded to silicon [32,33]. Low-intensity peaks at positions 228 and 231.4 eV corresponded to Mo-O bonds [34]. According to the B1s spectra, Mo-B bonds were also present in the coatings [35]. In the Y3d spectrum, peaks at positions 158.5 and 160.6 eV indicated the presence of Y–Si–O bonds in the coatings corresponding to yttrium silicate [36]. The doublet at 155.9 and 157.9 eV could be attributed to the metallic Y-Y bond [35,36]. The Si2p spectrum, with a maximum at position 102 eV, confirmed the presence of Y-Si-O bonds. A high-intensity peak at a binding energy of 99.36 eV corresponded to molybdenum silicide [35]. Thus, Mo-Si and Mo-B bonds were present in the coatings, and yttrium atoms were predominantly bonded to silicon and oxygen atoms.

*3.2. Oxidation Resistance*

Oxidation kinetics curves of the coatings at temperature of 1000 °C are shown in Figure 4a. The Mo-Si-B coating obtained by DCMS showed a specific mass change $\Delta m/S = -0.40$ mg/cm$^2$ at an exposure time of 10 to 300 min. Switching to the HIP-IMS mode contributed to a decrease in the $\Delta m/S$ parameter of the Mo-Si-B coating to $-0.09$ mg/cm$^2$. For the Mo-Y-Si-B DCMS sample, a maximum $\Delta m/S = -3.2$ mg/cm$^2$ was observed due to rapid oxidation, accompanied by intensive evaporation of MoOx and BOx [37,38]. The Mo-Y-Si-B coating obtained by the HIPIMS method was characterized by $\Delta m/S = -0.7$ mg/cm$^2$ at the maximum exposure time, which was 4.5 times lower

than the values obtained for the DCMS sample of the same composition. This effect can be associated with structure densification and a decrease in the defectiveness of coatings upon transition from DCMS to HIPIMS [20,39,40]. The Mo-Zr-Y-Si-B coating showed a consistently low $\Delta m/S = -0.16$ mg/cm$^2$ at exposures of 10–300 min. It can be concluded that the joint doping of the base coating with yttrium and zirconium contributed to an increase in the resistance to oxidation at a temperature of 1000 °C. The positive effect of the combined introduction of Y-Zr into the Nb-Ti-Al-Si silicide coatings was also revealed in [41]. The authors found that the high thermal stability of the coatings was associated with the formation of a protective film based on SiOx, the upper layer of which contains YZrOx particles.

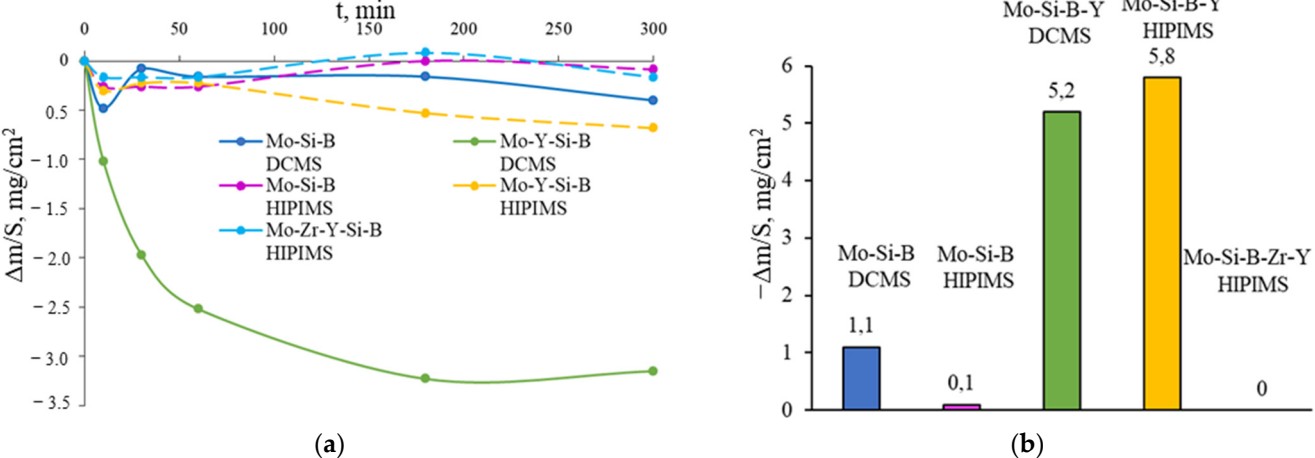

**Figure 4.** The dependence of the specific mass change on time at a temperature of 1000 °C (**a**) and the mass change as a result of annealing of the coatings at 1000 °C followed by cooling to 10 °C in water (**b**) for Mo-(Y,Zr)-Si-B coatings obtained by the DCMS and HIPIMS methods.

When visually evaluating the appearances of the samples at the end of the experiments, it was found that the Mo-Si-B DCMS coating slightly delaminated during the annealing process (Supplementary Materials). In this case, the delamination area occupied no more than 11% of the coating area after the end of the experiment. In the case of HIPIMS of the Mo-Si-B sample, no delamination was observed. Thus, the safety of Mo-Si-B coatings, which have a record oxidation resistance when heated to temperatures of 1500–1700 °C [5,7,42], can be increased using the HIPIMS mode. For the Mo-Y-Si-B coatings obtained by DCMS and HIPIMS, the first delamination was observed after only 10 min: the areas occupied by delamination were 50 and 10%, respectively. After a 300 min exposure, the DCMS coating was completely delaminated, while the HIPIMS Mo-Y-Si-B coating was retained by 55%. The decrease in adhesive strength under thermal action with the introduction of Y into the composition of Mo-Si-B coatings was associated with an increase in residual stresses from 0.5 to 1.4 GPa for DCMS samples and from 1.1 to 1.5 GPa for HIPIMS. Figure 5 shows two-dimensional profiles of the Mo-Si-B and Mo-Y-Si-B coatings obtained by the DCMS and HIPIMS methods.

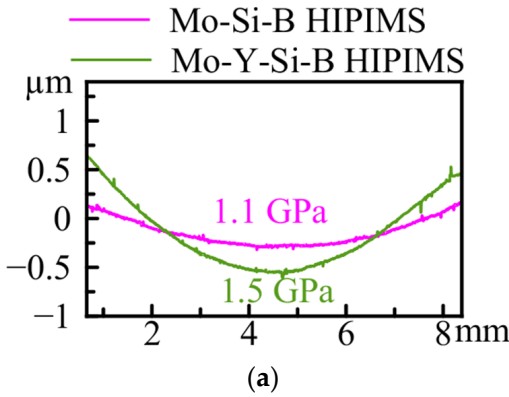
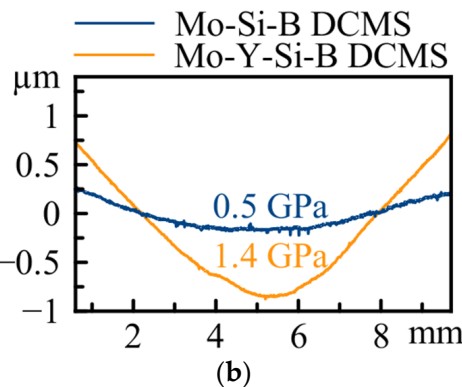

**Figure 5.** Two-dimensional profiles of samples obtained by the DCMS (**a**) and HIPIMS (**b**) methods.

No delamination was observed for the Mo-Zr-Y-Si-B HIPIMS coating during annealing. At 300 min, oxidized areas were observed on the surface of the sample, occupying about 45% of the coating area. It should be noted that the Mo-Zr-Y-Si-B coating, which had a minimum thickness of 3.2 μm, was characterized by greater protective properties compared to the Mo-Y-Si-B HIPIMS coating, which had a thickness of 6.3 μm.

Thus, the positive effect of increasing oxidation resistance by means of introducing yttrium into the composition of Mo-Si-B coatings is limited by high internal stresses, which, when heated, lead to the partial destruction of the coating. Additional doping of Mo-Y-Si-B coatings with zirconium makes it possible to improve the oxidation resistance and increase the protective properties of the base Mo-Si-B sample. The transition from DCMS to HIPIMS led to an increase in the oxidation resistance of Mo-Si-B and Mo-Y-Si-B coatings due to an increase in adhesive strength. The best oxidation resistance was characterized by HIPIMS coatings Mo-Si-B and Mo-Zr-Y-Si-B.

Single-stage annealing of the coatings at 1000 °C for 60 min, followed by cooling to 10 °C in water, showed that the Mo-Si-B DCMS coating was characterized by $\Delta m/S = -1.1$ mg/cm$^2$ (Figure 4b). The transition from the DCMS mode to HIPIMS led to a decrease in the $\Delta m/S$ parameter by a factor of 10. DCMS and HIPIMS Mo-Y-Si-B coatings showed close values of $\Delta m/S = -5.2$ and $-5.8$ mg/cm$^2$, respectively. A minimum specific weight change of $-0.01$ mg/cm$^2$ was observed for the HIPIMS coating of Mo-Zr-Y-Si-B. When evaluating the appearances of the coatings after testing, it was found that the DCMS and HIPIMS coatings of Mo-Si-B, as well as the HIPIMS coating of Mo-Zr-Y-Si-B, retained their integrity (Supplementary Materials). Mo-Y-Si-B coatings were completely exfoliated under conditions of a sharp temperature drop. Taking into account the specific weight change and the appearance of the samples, it can be argued that the HIPIMS coatings of Mo-Si-B and Mo-Zr-Y-Si-B showed the best resistance to destruction under conditions of a sharp temperature drop.

Cross-section and top-view SEM images of DCMS coatings after annealing at 1300 °C are shown in Figure 5. On the surface of the Mo-Si-B coating obtained using the DCMS method, the formation of a network of cracks was observed (Figure 6a). Note that cracking of the coating can adversely affect its oxidation resistance, since the presence of cracks facilitates the diffusion of oxygen deep into the material [40]. The formation of cracks during heating–cooling in coatings doped with yttrium was not observed (Figure 5b). This effect is associated with the suppression of columnar grain growth and refinement of the coating structure. Similar patterns were described earlier in the study on Mo-Si-B coatings doped with hafnium or zirconium [7,42]. It should be noted that for the Mo-Si-B coating, the transition from DCMS to HIPIMS mode due to structure modification also contributed to an increase in crack resistance at elevated temperatures.

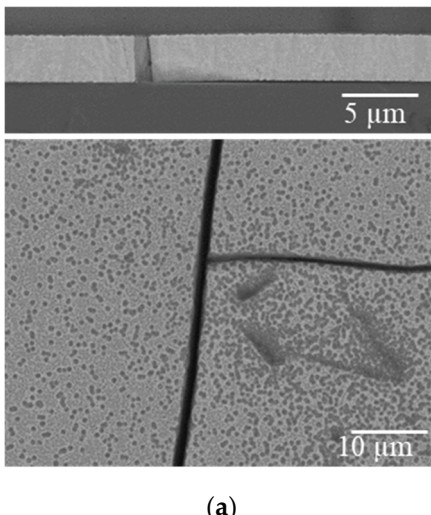 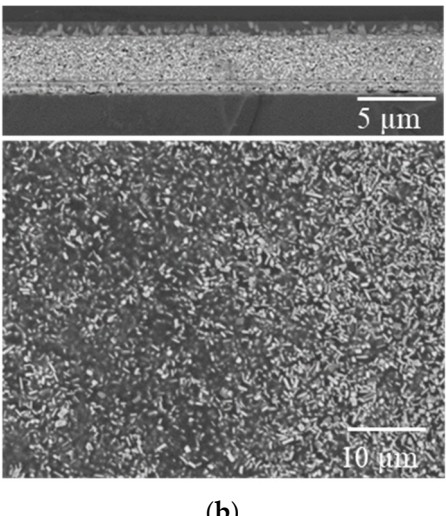

(**a**) (**b**)

**Figure 6.** Cross-section and top-view SEM images of Mo-Si-B (**a**) and Mo-Y-Si-B (**b**) DCMS coatings after annealing in air at 1300 °C.

On the surface of the Mo-Si-B coating obtained by the DCMS method, the formation of a network of cracks was observed (Figure 6a). Note that cracking of the coating can adversely affect its oxidation resistance, since the presence of cracks facilitates the diffusion of oxygen deep into the material [40]. The formation of cracks during heating–cooling in coatings doped with yttrium was not observed (Figure 6b). This effect was associated with the suppression of columnar grain growth and refinement of the coating structure. Similar patterns were described earlier in the study regarding Mo-Si-B coatings doped with hafnium or zirconium [7,42]. It should be noted that for the Mo-Si-B coating, the transition from the DCMS mode to HIPIMS due to structural modification also contributed to an increase in crack resistance at elevated temperatures.

According to the EDS data, after annealing at 1300 °C, an oxide layer based on SiOx with a thickness of ~500 nm was formed on the surface of the Mo-Si-B DCMS coating (Figure 7a). The SiOx surface layer, in the case of the Mo-Si-B HIPIMS coating, had a thickness of ~300 nm, which was 65% lower than that for the DCMS coating of the same composition (Figure 7b). In the case of the DCMS coating of Mo-Y-Si-B, $Y_2O_3$ particles with 0.1–1.5 μm in size were evenly distributed in the surface oxide layer based on SiOx with a thickness of ~2.0 μm (Figure 7a). Yttrium oxide crystallites also filled the pores in the non-oxidized layer of the Mo-Y-Si-B coating, thereby sealing defects and having a positive effect on the oxidation resistance of the coating. Previously, a similar effect was shown in [40]: the introduction of Hf into the composition of Mo-Si-B coatings led to an increase in oxidation resistance due to the formation of hafnium oxide particles in the pores of the non-oxidized coating layer.

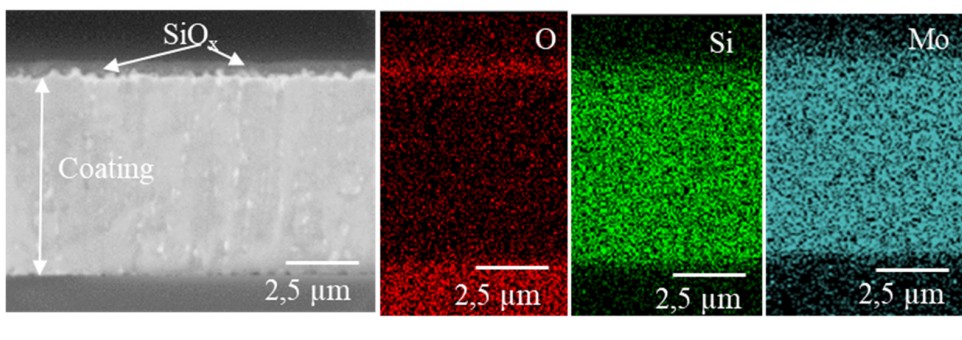

(**a**)

**Figure 7.** *Cont.*

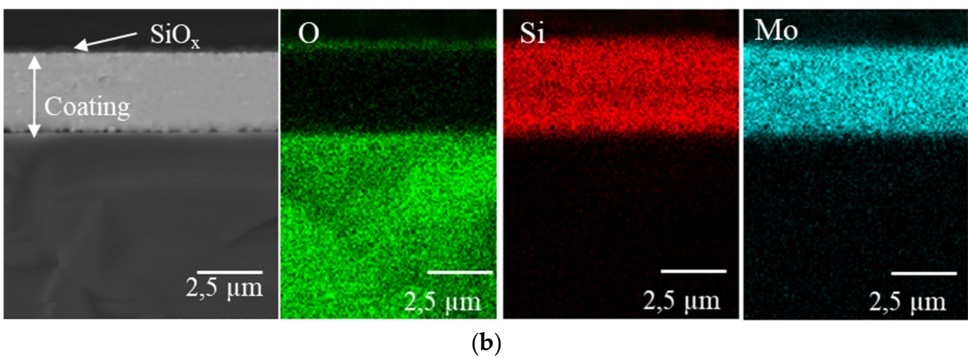

(**b**)

**Figure 7.** Cross-section SEM images and EDS maps for DCMS (**a**) and HIPIMS (**b**) Mo-Si–B coatings after annealing in air at 1300 °C.

In the case of the Mo-Y-Si-B HIPIMS coating, an oxide layer of SiOx+Y$_2$O$_3$, 1.5 μm thick, was formed on the surface during heating (Figure 8). In contrast to the DCMS sample, Y$_2$O$_3$ crystallites 0.3–1.5 μm in size were concentrated in the upper part of the oxide layer of the Mo-Y-Si-B HIPIMS coating. The transition from DCMS to HIPIMS contributed to a decrease of 25% in the thickness of the oxide layer.

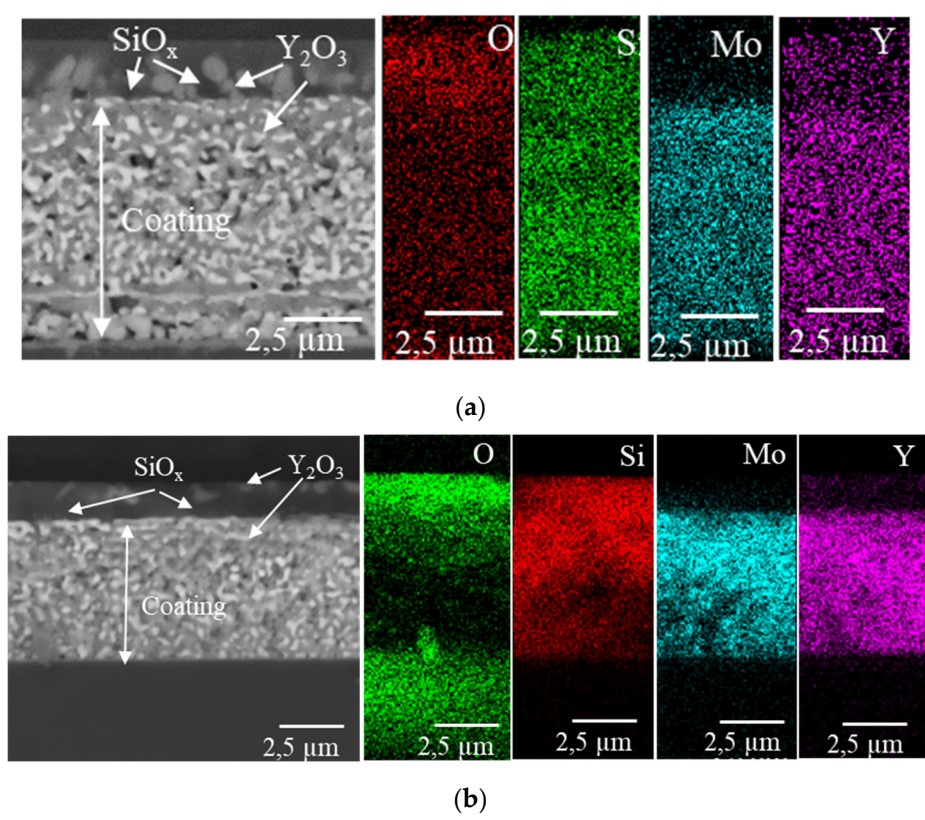

**Figure 8.** Cross-section SEM images and EDS maps for DCMS (**a**) and HIPIMS (**b**) Mo-Y-Si-B coatings after annealing in air at 1300 °C.

Thus, the introduction of yttrium into the composition of Mo-Si-B DCMS coatings led to an increase in crack resistance at high temperatures. The protective properties of the Mo-Y-Si-B coatings were provided by surface defect-free oxide films SiOx+Y$_2$O$_3$, which prevented the diffusion of oxygen deep into the coating. The transition from the DCMS mode to HIPIMS contributed to a decrease in the thickness of the oxide layer of 65 and 25% for the Mo-Si-B and Mo-Y-Si-B coatings, respectively.

X-ray diffraction patterns of Mo-Si-B and Mo-Y-Si-B coatings obtained by the DCMS and HIPIMS methods after annealing in air at 1300 °C are shown in Figure 9. Peaks corresponding to the material of the $Al_2O_3$ substrate were observed in the XRD patterns of all coatings. There were also low-intensity peaks corresponding to the (100) and (110) planes of the h-$MoSi_2$ phase, which were the main structural components of the coatings in the initial state. For the DCMS and HIPIMS Mo-Si-B coatings, the main high-intensity peaks were revealed at the positions of $2\theta$ = 22.6, 39.7, 46.2, 57.4 and 72.2°, corresponding to the planes (002), (110), (112), (200), and (006) of the t-$MoSi_2$ phase (ICDD 41-0612). The crystallite sizes in this phase, determined by the Scherrer formula from the most intense line (002), were 110 and 73 nm for the DCMS and HIPIMS coatings, respectively. Reflections were observed indicating the formation of the t-$Mo_5Si_3$ phase upon heating the coatings (ICDD 34-0371). The peaks observed at $2\theta$ = 18.5, 21.0, 26.0, 32.9 and 60.7° were attributed to the hexagonal h-$MoO_3$ phase (ICDD 65-0141). Upon shifting from DCMS to HIPIMS, the intensities of the lines of the h-$MoO_3$ phase increased.

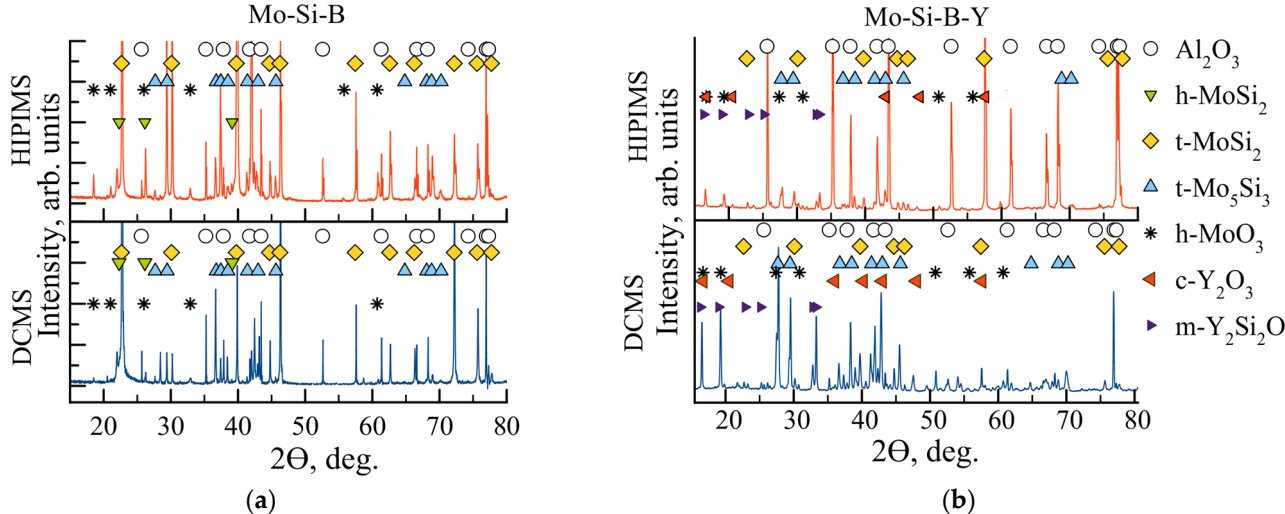

**Figure 9.** XRD patterns of the Mo-Si-B (**a**) and Mo-Y-Si-B (**b**) coatings after annealing in air at 1300 °C.

The DCMS and HIPIMS diffraction patterns of the Mo-Y-Si-B coatings revealed peaks in the t-$MoSi_2$ and t-$Mo_5Si_3$ phases corresponding to the non-oxidized layer (Figure 9b). In the case of the DCMS sample, the high-intensity peaks belonged to the t-$Mo_5Si_3$ phase, while the dominant phase for the HIPIMS coating was t-$MoSi_2$. Peaks related to the hexagonal h-$MoSi_2$ phase were not observed for either coating. Peaks from the h-$MoO_3$, c-$Y_2O_3$, and m-$Y_2Si_2O_7$ (ICDD 74-2163) phases present in the oxide layer were also observed on the XRD patterns. It is likely that yttrium partially dissolves in $SiO_2$ with the formation of m-$Y_2SiO_7$. The formation of $Y_2O_3$ and $Y_2SiO_7$ oxides was observed in [43] during the oxidation of Zr-Y-Si-B-C coatings. It should be noted that the intensities of the h-$MoO_3$, m-$Y_2SiO_7$, and c-$Y_2O_3$ peaks decreased upon shifting from DCMS to HIPIMS, which may be due to the thinner oxide layer thickness of the Mo-Y-Si-B HIPIMS coating. The Mo-Zr-Y-Si-B coating was completely oxidized, with the formation of a transparent oxide layer. X-ray diffraction analysis (not shown) revealed the presence of h-$MoO_3$, t-$ZrO_2$ (ICDD 75-9648), c-$Y_2O_3$, and m-$Y_2SiO_7$ phases.

Thus, the DCMS and HIPIMS Mo-Si-B coatings had high thermal stability: the h-$MoSi_2$ phase observed in the initial state was retained in the coatings at up to 1300 °C. The introduction of yttrium into the composition of the Mo-Si-B coatings led to some decrease in thermal stability: after the h-$MoSi_2$ → t-$MoSi_2$ phase transition, no traces of the h-$MoSi_2$ phase remained. The transition from DCMS to HIPIMS had no effect on the thermal stability of the coatings.

TEM micrographs and EDS maps for the Mo-Si-B DCMS coating annealed at 1300 °C are shown in Figure 10. On the EDS maps, it is possible to distinguish areas corresponding to the $Al_2O_3$ substrate, coating (MoSix), and oxide layer (SiOx). The phases were identified by analyzing the electron diffraction patterns taken from each particle. According to bright-field images of the structure and electron diffraction patterns, grains with interplanar distances of 0.345, 0.230, and 0.156 nm, corresponding to the rhombohedral $Al_2O_3$ phase (substrate material) with d/$n$ = 0.347, 0.237, and 0.160 nm, were observed in the Mo-Si-B DCMS coating sample on an $Al_2O_3$ substrate after annealing. An amorphous component, identified as $SiO_2$, was also detected. An electron diffraction pattern taken from a dark grain ~400 nm in size revealed point reflections with an interplanar spacing, d/$n$ = 0.202 nm.

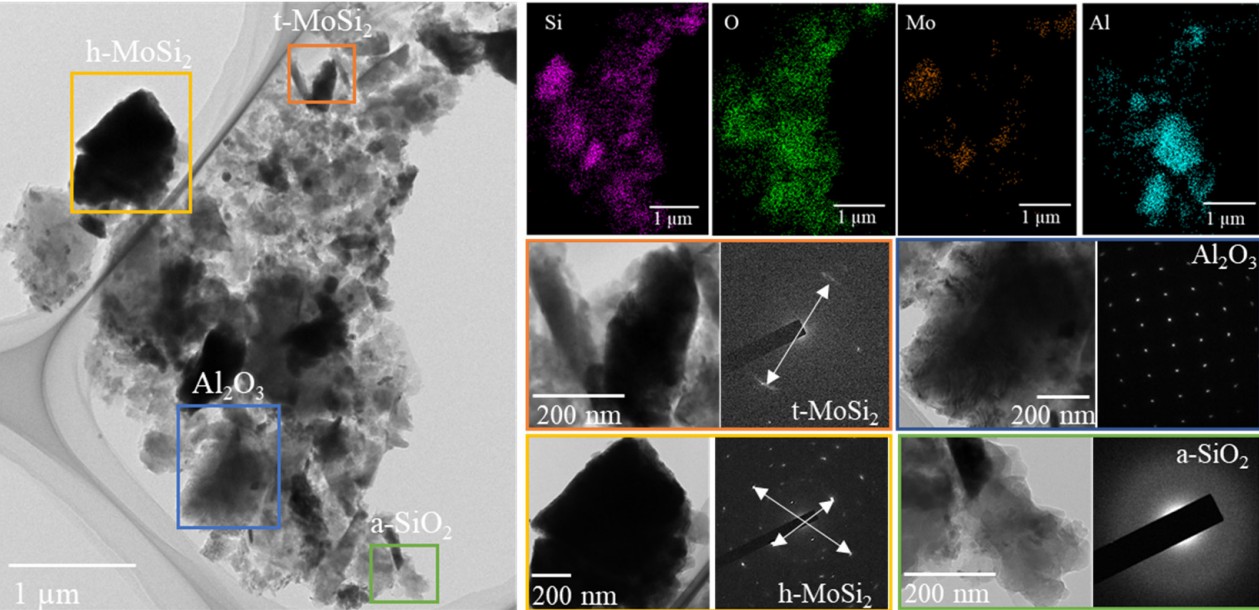

**Figure 10.** EDS maps, bright-field images of the structure, and SAED patterns of the Mo-Si-B DCMS coating after annealing at a temperature of 1300 °C.

These values are close to tabular data, d/$n$ = 0.203 nm, for the most intense line (103) of the $t$-$MoSi_2$ tetragonal phase. The interplanar distances determined according to an electron diffraction pattern taken from a particle with a size of ~1 μm were 0.390, 0.335, and 0.218 nm. These values are close to d/$n$ = 0.398, 0.340, and 0.217 nm for the $h$-$MoSi_2$ hexagonal phase. The presence of $t$-$MoSi_2$ and $h$-$MoSi_2$ phases in the annealed Mo-Si-B coating was also confirmed by XRD results.

For the Mo-Si-B HIPIMS coating, in addition to particles corresponding to the $Al_2O_3$ substrate material and amorphous $SiO_2$ inclusions, zones with high silicon and molybdenum content were revealed (Figure 11).

The electron diffraction patterns obtained from these areas showed the presence of reflections with d/$n$ = 0.296, 0.228, and 0.204 nm, corresponding to the most intense reflections ((101), (110), and (103)) of the $t$-$MoSi_2$ tetragonal phase and d/$n$ = 0.340, 0.257, and 0.221 nm, which can be identified as $h$-$MoSi_2$.

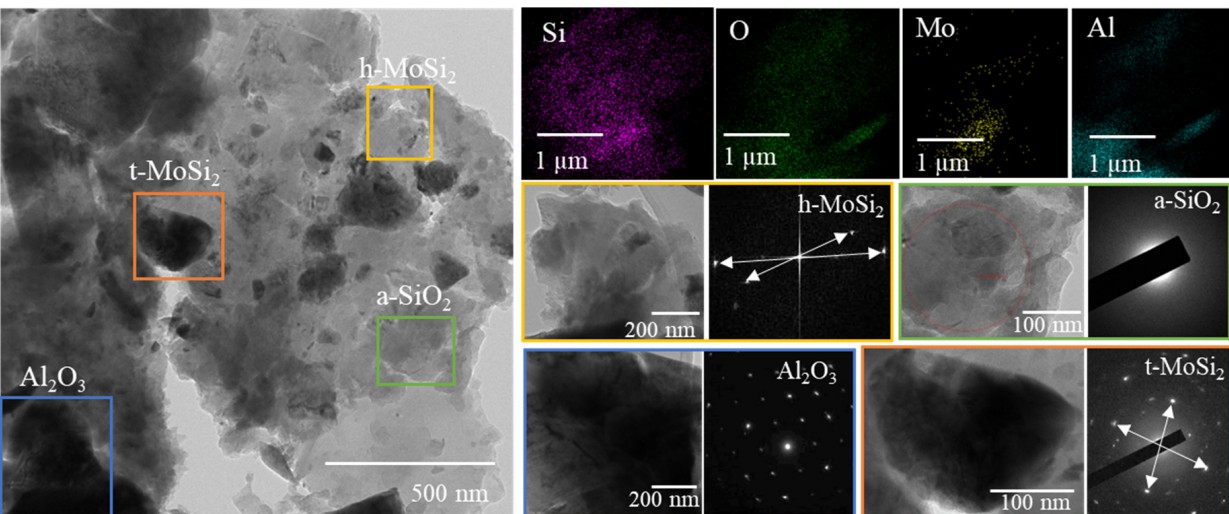

**Figure 11.** EDS maps, bright-field images of the structure, and SAED patterns of the Mo-Si-B HIPIMS coating after annealing at a temperature of 1300 °C.

For the Mo-Y-Si-B coating obtained by the DCMS method, according to the EMF maps, the layer structure contained particles of $Al_2O_3$, $MoSi_2$, and $Y_2O_3$, as well as regions corresponding to $SiO_2$ (Figure 12).

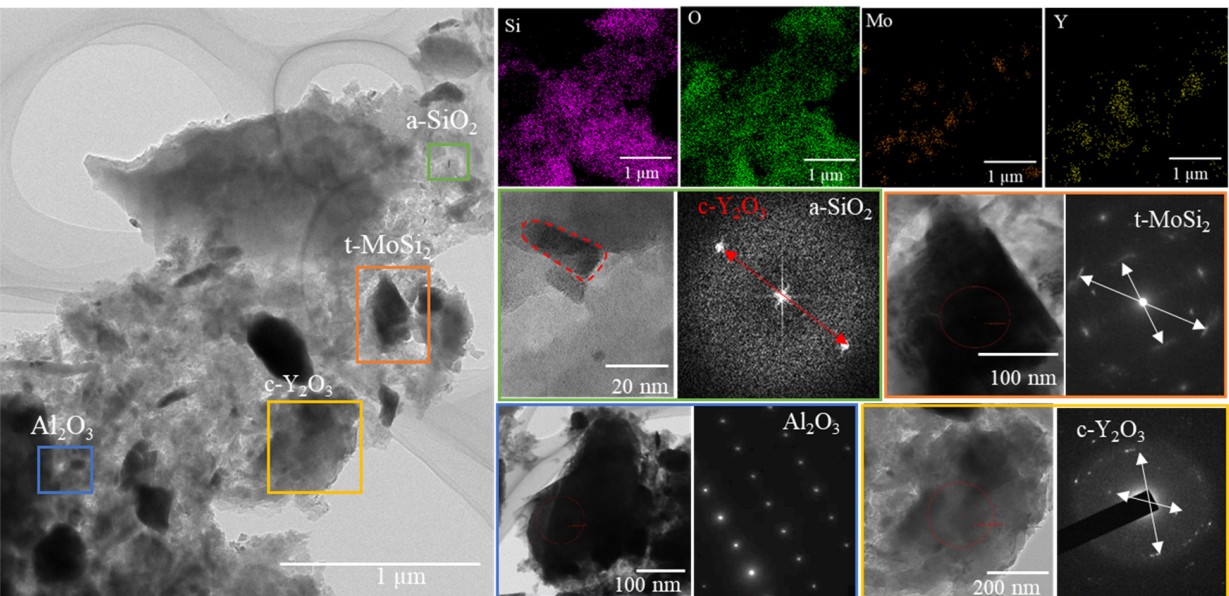

**Figure 12.** EDS maps, bright-field images of the structure, and SAED patterns of the Mo-Y-Si-B DCMS coating after annealing at a temperature of 1300 °C.

The results showed that the $SiO_2$ oxide layer was characterized by an amorphous structure, as evidenced by a wide ring on the FFT. Point reflections on the FFT with an interplanar spacing of 0.421 nm were attributed to the c-$Y_2O_3$ cubic phase with $d/n = 0.432$ nm. The electron diffraction patterns taken from yttrium-containing areas revealed narrow reflections with interplanar spacings of 0.512, 0.305, and 0.266 nm, close to the values of $d/n = 0.529$, 0.305, and 0.268 nm of the c-$Y_2O_3$ cubic phase. The reflections on the electron diffraction patterns of molybdenum-containing regions with $d/n = 0.296$, 0.228, and 0.204 nm corresponded to the most intense lines ((101), (110), and (103)) of the t-$MoSi_2$ tetragonal phase.

Thus, TEM studies carried out after annealing at a temperature of 1300 °C confirmed the presence of an h-MoSi$_2$ phase in the DCMS and HIPIMS Mo-Si-B coatings. The study of the fine structure of the oxidized Mo-Y-Si-B coatings also made it possible to more accurately determine the stoichiometry of the yttrium oxide phase c-Y$_2$O$_3$.

## 4. Conclusions

Coatings in the Mo-(Y,Zr)-Si-B system were obtained by DCMS and HIPIMS using mosaic targets.

Structural studies showed that the Mo-Si-B coatings were characterized by a crystal structure with a crystallite size of 50 nm for the main h-MoSi$_2$ phase. The introduction of Y and Zr into the composition of Mo-Si-B coatings led to a decrease in the size of the h-MoSi$_2$ crystallites by a factor of 10 and an increase in the proportion of the amorphous component with the formation of a nanocomposite structure. It has been established that in the case of Mo-Y-Si-B coatings, yttrium atoms form Y-Y and Y-Si-O chemical bonds. Complex alloying with Y and Zr contributed to the increase in the oxidation resistance of Mo-Si-B samples at a temperature of 1000 °C and their resistance to destruction under conditions of a sharp temperature drop (1000→10 °C). The introduction of yttrium into the composition of Mo-Si-B coatings reduced the tendency of Mo-Si-B coatings to form cracks when heated to a temperature of 1300 °C. The high oxidation resistance of yttrium-containing coatings was achieved by forming a dense oxide layer of a-SiO$_2$+h-MoO$_3$ containing c-Y$_2$O$_3$ crystallites 0.1–1.5 μm in size.

The transition from the DCMS to HIPIMS mode led to suppression of the columnar growth of the Mo-Si-B coating. For Mo-Y-Si-B nanocomposite coatings, no structural changes were revealed when the sputtering mode was changed. The use of HIPIMS in the case of Mo-Y-Si-B coatings led to a decrease in the oxidation rate at T = 1000 °C by 4.5 times, as well as in the thickness of the oxide layer at a temperature of 1300 °C by ~2 times.

**Supplementary Materials:** The following supporting information can be downloaded at: https://www.mdpi.com/article/10.3390/jcs7050185/s1, Figure S1: Appearance of the coatings after annealing at a temperature of 1000 °C and exposures of 10–300 min, and after annealing at 1000 °C followed by cooling to 20 °C and high-temperature annealing at 1300 °C.

**Author Contributions:** Supervision, P.V.K.-K.; conceptualization, A.D.S. and P.V.K.-K.; formal analysis, A.D.S. and P.V.K.-K.; investigation, A.D.S., P.A.L., A.V.N. and P.V.K.-K.; resources, E.A.L.; writing—original draft preparation, A.D.S.; writing—review and editing, P.V.K.-K. All authors have read and agreed to the published version of the manuscript.

**Funding:** This work was performed with financial support from the Ministry of Science and Higher Education of the Russian Federation (State Assignment project No. 0718-2020-0034).

**Data Availability Statement:** Not applicable.

**Acknowledgments:** The authors are grateful to N.V. Shvyndina for the SEM-EDS study and A.V. Bondarev for help with the XPS study.

**Conflicts of Interest:** The authors declare no conflict of interest.

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
