# Peer review of "Structure and Oxidation Resistance of Mo-Y-Zr-Si-B Coatings Deposited by DCMS and HIPIMS Methods Using Mosaic Targets"

_jcs, doi:10.3390/jcs7050185_

Round 1
Reviewer 1 Report
Review of jcs-2256186, entitled "Structure and oxidation resistance of Mo-Y-Zr-Si-B coatings deposited by DCMS and HIPIMS methods using mosaic targets":
This manuscript reports the Mo-(Y,Zr)-Si-B coatings with excellent oxidation resistance deposited by DCMS and HIPIMS. The microstructure, bonding characteristic and phase constitution of the as-sputtered coatings were systematically investigated. Air oxidation test was conducted, and the oxides were discussed. The manuscript is well arranged, it is recommended to be accepted for publication after revising the following aspects:
1. For the morphologies of DCMS deposited films, it is reasonable to speculate that columnar grains could be obtained. While it is hard to accept it from results in Figure 1.
2. Y-Si-O was found in the as deposited Mo-Y-Si-B coating, which is a bit strange for high vacuum conditions.
3. In Figure 4a, Mo-Zr-Y-Si-B coating exhibits the largest weight loss. How about the effect of 22 at.% B.
4. For figures 9, 10 and 11, it would be great helpful for understand results with labeling the initial surface of the film.
5. Please check with the CTE value of MoSi2 in the introduction.
6. In Figure 8, the oxide products require identification. It is widely accepted that cubic Y2O3 instead of fcc. Either Y2SiO5 or Y2Si2O7 is the stable phase rather than Y2SiO7.
7. It would be great to combine the single-sentence paragraphs with other ones. For instance, line 151, line 183 and 184, as well as that on page 9.
Reviewer 2 Report
Mo-(Y,Zr)-Si-B coatings were fabricated by DCMS and HIPIMS. The main result of this article is that the Mo-Y-Si-B by HIPIMS case has much disadvantage of oxidation resistance. Author argued that the main reason of this property is due to the fact that the delamination results in destruction of the coating. I have seen discussions, however, cannot find scientific reasoning of the results. TEM, SEM, XRD data looks nice, but there are no meaningful reasoning. Analysis data and the oxidation resistance properties were not scientifically connected. Many works, but not organized. This author should clear the reason why the delamination happened in the case of Y addition.
Reviewer 3 Report
This manuscript by Sytchenko et. al, titled “Structure and oxidation resistance of Mo-Y-Zr-Si-B coatings deposited by DCMS and HIPIMS methods using mosaic targets” presents a very interesting work on improving the oxidation resistance of Mo-Si-B coatings. The manuscript has been very well written with detailed work which clearly presents the current research gap and a viable solution towards it. However, the manuscript lacks a couple of important points that have been mentioned below:
· Please check for mistypes and/or minor grammatical errors. For example: An incomplete sentence on Page 1, Like 38, “At the same time, at a… oxidation of Mo”. Mistype on Page 2, Line 48, “The introduction of Y2O3 into…”
· Page 2, Line 83, “The coatings were deposited on functionally graded targets…”: The authors need to make a separate paragraph explaining how the targets were fabricated by hot pressing and what led to the decision of making these compositions of 90%MoSi2+10%MoB, 80%(90%MoSi2+10%MoB)+20%ZrB2. This will immensely help the readers and researchers in repeatability of this work.
· Page 4, Table 1: Are the elemental compositions obtained from EDS data and how many Point of Interests were noted? The authors need to provide the standard deviation from a statistically significant data set.
· Page 6, Figure 3: Distinguishable colors and bolded lines in the plots can help in understanding and interpretation better.
Round 2
Reviewer 1 Report
All the comments have been properly revised. Please check the sentences describing grains size of Mo, Si, zirconium as well as in Materials and Methods.
Reviewer 2 Report
Dear Authors.
Thank you for your revision.
I still have no doubt that the main theme of this article is the reasoning or clear discussion why the addition of Y in Mo-Si-B give disadvantage for the system. I write down 2 opinions below;
(1) You added 2D profiles of coating layer for each systems. The internal stress increased from 0.5 to 1.4 for DCMS and 1.1 to 1.5 for HIPIMS respectively. How could you calculate the internal stress from this profiles? You should put these 2D profiles into manuscript.
(2) Y addition into system do not have merit in oxidation properties. So there are no improvement in properties. This is serious weak point for publishing your work.
Best regard,
Reviewer
Reviewer 3 Report
The authors have made significant changes as suggested, the manuscript looks ready for publication.
Author Response
Thank you for your positive response on our manuscript.
Round 3
Reviewer 1 Report
All the comments are properly responded. It is recommended to accept for publication.
Reviewer 2 Report
Residual stresses were measured by curvature profiles of each samples. It increase as Y added in both DCMS and HIPIMS methods. Y addition give good oxidation resistance but increase residual stress and resulting in the increasing delamination. The effects of Y addition were scientifically discussed and showed the supporting results.